# Neoadjuvant PD-1 blockade in surgically resectable desmoplastic melanoma: cohort A of the phase 2 SWOG S1512 trial

Kari L. Kendra [1,17] ✉, Shay L. Bellasea[2,3], Zeynep Eroglu[4], Siwen Hu-Lieskovan[5], Katie M. Campbell [6], William E. Carson III[1], David A. Wada[5], Jose A. Plaza[1], Gino K. In[7], Alexandra Ikeguchi [8,13], John Hyngstrom[5,14], Andrew S. Brohl [4], Bartosz Chmielowski [6], Nikhil I. Khushalani[4], Joseph Markowitz[4], Marcus Monroe[5], Carlo M. Contreras[1], Tawnya Bowles[9], Kurt Norman[10], Egmidio Medina [6], Cynthia R. Gonzalez[6], Ignacio Baselga-Carretero[6], Ivan Perez Garcilazo[6], Agustin Vega-Crespo[6], Jia Ming Chen [6], Nataly Naser Al Deen[6], Sapna P. Patel [11], Kenneth F. Grossmann[5,15], Vernon K. Sondak[4], Elad Sharon[12,16], James Moon[2,3], Michael C. Wu [2,3] & Antoni Ribas [6,17] ✉

The phase 2 SWOG S1512 trial (NCT02775851) was designed to evaluate the response to pembrolizumab (anti-PD-1) in individuals with desmoplastic melanoma. Here we report the results of cohort A of the trial, evaluating the pathological complete response (pCR) rate of neoadjuvant PD-1 blockade in surgically resectable desmoplastic melanoma. Secondary endpoints included clinical response rate, overall survival and toxicities. Twenty-eight eligible individuals with resectable desmoplastic melanoma received intravenous pembrolizumab (200 mg) every 3 weeks three times, followed by excision. Tissue samples before treatment, at 3–5 weeks after treatment initiation and at the time of surgery were reviewed. The primary endpoint of pCR rate by local pathological review was 71% (95% confidence interval, 51–87%; P < 0.001), which met the prespecified endpoint. There were two (7%) grade 3 treatment-related adverse events. At three years of follow-up, four participants have died, none known to be from melanoma or adverse events. In conclusion, neoadjuvant pembrolizumab in individuals with resectable desmoplastic melanoma results in a high pCR rate with acceptable safety profile. Clinicaltrials.gov: NCT02775851.

Desmoplastic melanoma is a rare subtype of melanoma that is usually amelanotic and arises from highly sun-exposed areas. Histologically, it is characterized by spindle cells in a background of abundant collagen and lymphoid cell aggregates. Genetically, desmoplastic melanoma is among the most highly mutated cancers, with the mutation pattern implicating ultraviolet (UV) light radiation as the dominant carcinogen. It harbors common NF1 loss-of-function mutations while it lacks common cutaneous melanoma oncogene driver mutations in BRAF or

NRAS[1–3]. At diagnosis, deep cancer cell infiltration and neurotropism are common. Current management for localized disease is surgical excision with or without radiation therapy and adjuvant anti-PD-1 (nivolumab or pembrolizumab) immunotherapy if stage IIB or higher[4]. Lesions frequently extend well beyond what is seen clinically or by imaging, particularly with the presence of perineural invasion. This frequently results in difficulty in achieving clear margins with surgical resection. Multiple surgeries and large excisions are often required, leading to

clinical defects in highly visible areas. Surgical morbidity is particularly challenging for older and frail individuals because desmoplastic melanoma is frequently associated with age.

Until recently, advanced desmoplastic melanoma was considered resistant to most systemic therapies, but a retrospective series demonstrated that this cancer is highly responsive to PD-1 blockade therapy[2]. This review of over 1,000 cases of advanced melanoma treated with anti-PD-1/PD-L1 therapy in early clinical trials resulted in the identification of 60 individuals with advanced desmoplastic melanoma. Anti-PD-1/PD-L1 therapy resulted in an objective response rate of 70%, with a clinical complete response rate of 32% (ref. [2]). Responses were noted in both pure and mixed histological subtypes of desmoplastic melanoma. This study suggested that advanced desmoplastic melanoma may be among the metastatic cancers with the highest response rates to single-agent immune checkpoint blockade (ICB) therapy[5].

There are potential advantages to neoadjuvant therapy in locally advanced desmoplastic melanoma. A high response rate in the neoadjuvant setting could change the course of therapy for individuals with locally advanced disease, with improved local and systemic results. If substantial tumor regression was achieved, the use of radical surgery and radiation therapy may be reduced along with surgery-related morbidities, resulting in a higher quality of life for patients. This is particularly relevant for cases with positive surgical margins at the wide surgical excision due to the frequent deep perineural invasion in primary desmoplastic melanomas. Therefore, with the knowledge that neoadjuvant systemic anti-PD-1 therapy achieves high pathological complete response (pCR) rates, patients may be spared from repeated surgeries for residual local disease. Many primary desmoplastic melanomas are diagnosed at T3b or T4 stage, hence being a candidate for one year of adjuvant anti-PD-1 therapy; three doses of neoadjuvant anti-PD-1 therapy could represent an improvement in cost and lower the chances of experiencing toxicities if not followed by the full adjuvant anti-PD-1 therapy. Furthermore, results from the SWOG S1801 randomized clinical trial comparing neoadjuvant–adjuvant to adjuvant-only pembrolizumab provide the evidence that three doses of neoadjuvant pembrolizumab, followed by 15 doses of adjuvant pembrolizumab, improved the event-free survival of patients with locally advanced melanoma over using 18 doses of pembrolizumab in the adjuvant setting only[6]. Based on these data, patients with desmoplastic melanoma would be expected to have a lower frequency of systemic relapses when receiving anti-PD-1 in the neoadjuvant setting, likely by reinvigorating tumor-infiltrating antitumor T cells with PD-1 blockade before surgical excision[7].

SWOG S1512 was a prospective clinical trial with two cohorts of patients with desmoplastic melanoma treated with single-agent pembrolizumab: (1) cohort A enrolled patients with locally advanced disease receiving neoadjuvant therapy and (2) cohort B enrolled patients with advanced desmoplastic melanoma not deemed surgically resectable (published in ref. [8]). In cohort A, reported here, we tested the hypothesis that the administration of single-agent anti-PD-1 with pembrolizumab given in the neoadjuvant setting would have a high rate of pCR and be safe in patients with locally advanced desmoplastic melanoma, based on the favorable features for response to anti-PD-1 of a high mutational load resulting from chronic sun damage and pre-existing lymphoid cell aggregates.

## Results

### Patients and treatment

Between July 2017 and May 2021, 30 patients with resectable desmoplastic melanoma (American Joint Committee on Cancer Eighth Edition), stages I–III, were registered to the study and enrolled at ten investigator sites across the United States. The primary endpoint was the rate of pCR; a sample size ($n = 25$) was based on a single-stage design with 90% power to rule out a pCR rate of 5% (null pCR rate) at the 3.4% level, if the true pCR rate was 25% (alternative pCR rate). Secondary endpoints

were clinical response rate, overall survival (OS), relapse-free survival (RFS) and toxicities. Melanoma-specific survival was a post hoc endpoint. One patient refused protocol therapy and withdrew consent, and one patient was deemed ineligible after a review of the pathology report indicated that their disease was not consistent with desmoplastic melanoma; 28 patients were eligible, received protocol therapy and were considered the intent-to-treat population for the assessment of the primary endpoint of pCR rate (Fig. 1). Patients had a median age 75 years (range 37–91), 21 patients were men (75%) and 27 patients were white (96%), consistent with the known epidemiology of this cancer. The most common primary site of disease was the head and neck area (19 patients, 68%). Twenty-three (82%) of the patients were enrolled with desmoplastic melanoma at the time of their initial, primary diagnosis, and four patients (14%) were enrolled with locally recurrent disease (Table 1 and Supplementary Table 1). Five of the 28 enrolled patients (18%) had node-positive disease detected by either clinical exam or computed tomography (CT) scan at the time of enrollment (Table 1 and Supplementary Table 1). As of 20 May 2024, the median follow-up from date of registration was 42 months (range 2–64) for all patients alive. The protocol called for three infusions of pembrolizumab every 3 weeks before surgery at 9 weeks, with the possibility of receiving a fourth neoadjuvant infusion if there was no clinical response and the option of receiving adjuvant pembrolizumab up to 15 infusions after surgery (Fig. 2). There was an optional on-therapy biopsy at 3 to 5 weeks from the start of neoadjuvant therapy. Twenty-five (89%) patients received the scheduled three cycles of neoadjuvant pembrolizumab. One patient discontinued therapy after cycle 1 due to colitis. Two patients received an additional fourth dose of neoadjuvant pembrolizumab (Fig. 2 and Supplementary Table 2). Median time from cycle 1, day 1 to surgery was 80 days (range 52–135). None of the 28 patients in this cohort received adjuvant pembrolizumab after surgery.

### Clinical response

The secondary endpoint of clinical responses defined by Response Evaluation Criteria in Solid Tumors (RECIST) version 1.1 criteria was assessed using either radiographic measurements from imaging scans or clinical measurements of the tumor documented by photographs (Fig. 3a; the authors affirm that human research participants provided written consent for the publication of the images). Of the 28 participants, three had nonmeasurable disease at baseline and were not included in the analysis of clinical response (Fig. 1a and Extended Data Fig. 1). Of the 25 patients with measurable disease at baseline, 15 were measured by radiographic assessment and 10 were measured by clinical assessment. At the 9-week clinical response assessment time, four (16%) patients had a CR and eight (32%) had a partial response (PR), for an overall clinical response rate of 48% (95% confidence interval (95% CI), 28–69%; Supplementary Tables 2 and 3). Stable disease was documented in nine patients (36%), and one patient (4%) showed disease progression by clinical assessment. One patient did not have a 9-week assessment before surgery, and two patients were not assessed using the same technique as at baseline; all three were considered inadequately assessed and counted in the denominator as having no response to therapy for the assessment of clinical responses. Neither of the two patients who received a fourth infusion of neoadjuvant pembrolizumab did so due to disease progression by RECIST criteria. PT0502 had surgery scheduling issues and received a fourth infusion with a major clinical and pathological response. PT0785 had restaging CT scans after three infusions of neoadjuvant pembrolizumab with evidence of a clinical PR at the measurable disease at the primary site (from 3.8 cm to 2.3 cm) and axillary metastases (from 2.6 cm to 1.8 cm), but there were questions about nonmeasurable lymph nodes with a mild increase in size. Given the uncertainty, the patient was given a fourth infusion followed by surgical resection, which showed a pathological response in the skin primary site but the presence of melanoma in 2 of 30 resected lymph nodes. In summary, 21 of 25 patients with assessable disease (84%) had no evidence of clinical progression during the administration of

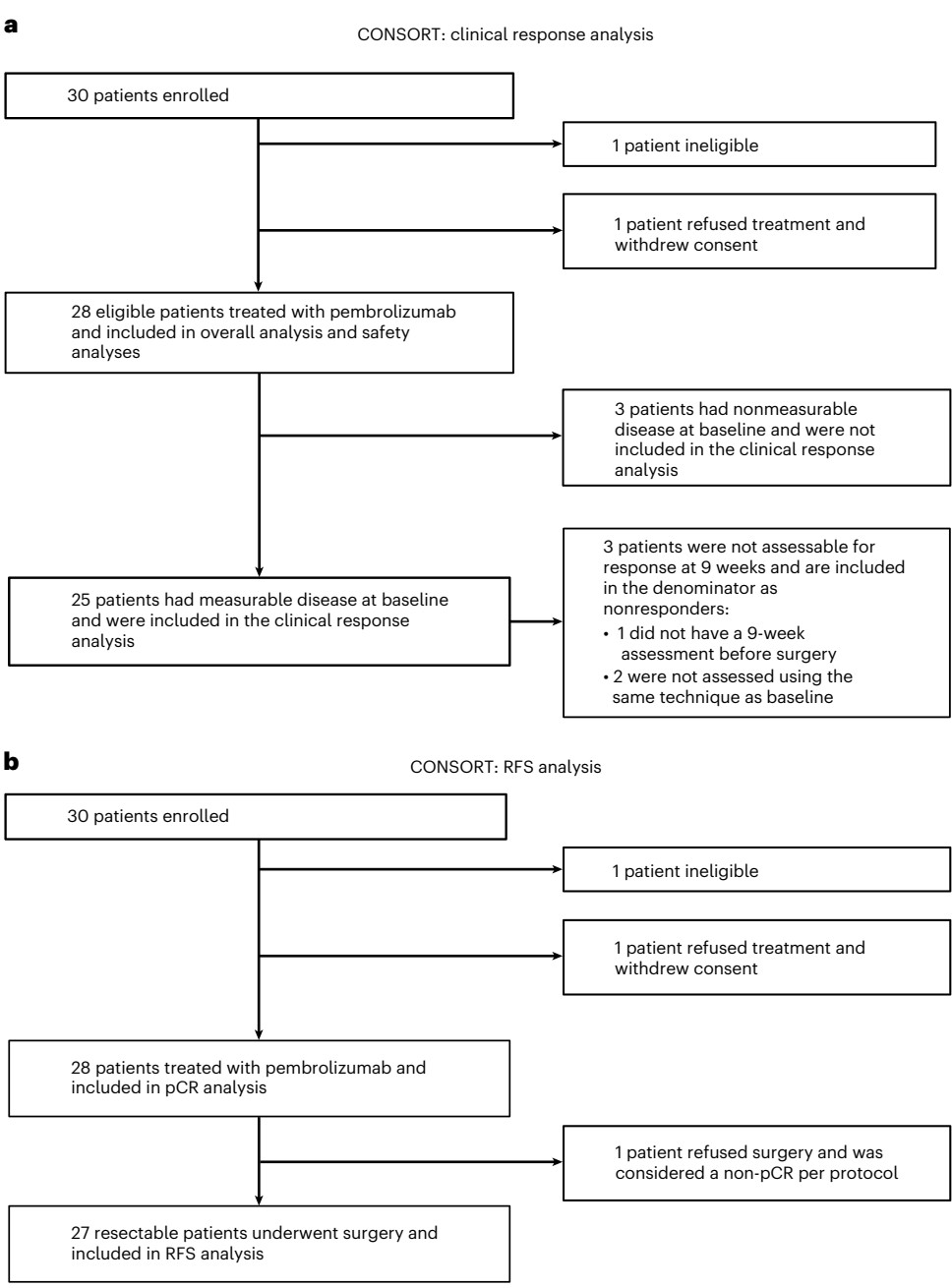

**Fig. 1 | CONSORT diagram.** Clinical trial S1512, cohort A, where neoadjuvant pembrolizumab was administered to patients with resectable desmoplastic melanoma. **a**, Clinical response analysis. **b**, RFS analysis.

neoadjuvant pembrolizumab (Figs. 3b and 4, Extended Data Fig. 2 and Supplementary Table 2).

### Surgical resection
Of the 28 eligible patients who began neoadjuvant pembrolizumab, 27 underwent surgical resection of their tumor (Fig. 1 and Extended Data Fig. 1). One patient (4%) elected not to proceed with surgical resection and was considered as not having a pCR per protocol. One patient (4%) discontinued pembrolizumab early due to colitis but was able to successfully undergo surgical resection. Twenty-six (93%) patients underwent wide excision, of which 18 (69%) underwent a sentinel lymph node biopsy and two (8%) underwent a lymph node dissection. One patient (4%) underwent surgical resection of a nodal recurrence without wide excision of the primary skin site as the lymph node was the only site of disease. Two patients had subsequent surgical

resections to obtain clear margins. The five patients with lymph node involvement at baseline all underwent lymphadenectomies after neoadjuvant pembrolizumab (Supplementary Table 4).

### Pathological response rate by local institutional assessment
The primary endpoint to assess pCR rate was based on local pathology evaluation of the resected specimen after neoadjuvant pembrolizumab in the 28 eligible patients (Fig. 1 and Extended Data Fig. 2). A pCR was reported in resection specimens of 20 of 28 patients (71%; 95% CI, 51–87%, $P < 0.001$; Supplementary Table 2) in the intent-to-treat population. Thus, the study successfully met its primary endpoint of demonstrating that neoadjuvant pembrolizumab results in a pCR rate that rules out the null hypothesis of a 25% pCR rate, which is the expectation for neoadjuvant single-agent anti-PD-1 treatment in cutaneous melanoma not of the desmoplastic subtype.

## Table 1 | Patient and disease characteristics

| Characteristic | (*N*=28) | |
|---|---|---|
| Age (years) | 75 | (37, 91) |
| **Sex** | | |
| Men | 21 | (75%) |
| Women | 7 | (25%) |
| **Race** | | |
| White | 27 | (96%) |
| Unknown | 1 | (4%) |
| **Ethnicity** | | |
| Hispanic | 1 | (4%) |
| Not Hispanic | 25 | (89%) |
| Unknown | 2 | (7%) |
| **Performance status** | | |
| 0 | 22 | (79%) |
| 1 | 6 | (21%) |
| **Primary site of disease** | | |
| Head or neck | 19 | (68%) |
| Extremity | 5 | (18%) |
| Torso | 4 | (14%) |
| **AJCC 8th edition T classification[a]** | | |
| T1a | 1 | (4%) |
| T1b | 2 | (7%) |
| T2a | 6 | (21%) |
| T3a | 7 | (25%) |
| T3b | 1 | (4%) |
| T4a | 8 | (29%) |
| T4b | 2 | (7%) |
| TX | 1 | (4%) |
| **AJCC 8th edition N classification** | | |
| N0 | 23 | (82%) |
| N1b | 3 | (11%) |
| N2c | 2 | (7%) |
| **Disease status** | | |
| Primary | 23 | (82%) |
| Recurrent | 5 | (18%) |
| **LDH at baseline** | | |
| Elevated LDH | 3 | (11%) |
| Normal LDH | 25 | (89%) |

Patient characteristics among enrolled participants. Median (range) and number (percentage) are reported. Percentages may not sum to 100 because of rounding. [a]Minimal T stage based on the biopsy obtained. AJCC, American Joint Committee on Cancer Eighth Edition; LDH, lactate dehydrogenase.

### Central evaluation of desmoplastic melanoma diagnosis and pathological response

We undertook a retrospective centralized blinded pathology review of desmoplastic melanoma diagnosis and pathological response assessment by two independent pathologists (D.A.W. and J.A.P.). This was a preplanned, exploratory endpoint. Diagnostic pathology for central review was intended to be performed using baseline biopsies or archival biopsies obtained before treatment, at the time of the initial diagnosis of desmoplastic melanoma. When pretreatment samples were not available for central review, the on-treatment 3- to 5-week biopsy or

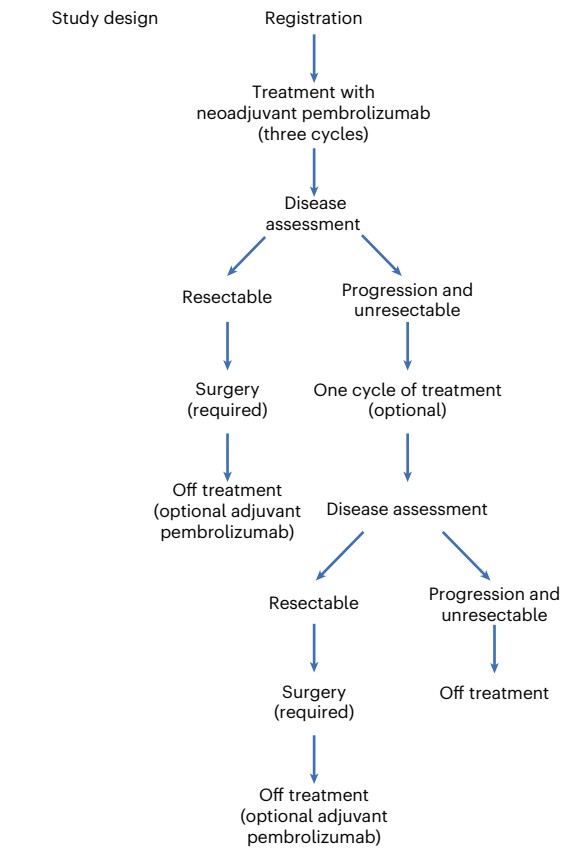

**Fig. 2 | Study schema.** Study design and treatment plan.

the surgical resection specimen was evaluated. Diagnostic pathology samples were not submitted from two cases, and from three cases the submitted samples did not contain tumor upon central pathology review (Supplementary Tables 5 and 6 and Extended Data Fig. 1), likely representing an error in submitting the sample, probably being from one of multiple tissue blocks that happened to not contain desmoplastic melanoma. Among the 23 cases with samples evaluable for central pathology review, 21 of the pretreatment biopsies were confirmed to be diagnostic of desmoplastic melanoma. Among the two pretreatment samples that were not consistent with desmoplastic melanoma in a diagnostic baseline sample, one participant (PT0289) was confirmed to have desmoplastic melanoma in the on-therapy biopsy, and the other participant (PT0771) had an on-therapy biopsy that had evidence of a pathological tumor response and had a pCR in the surgical resection.

The central pathology review of response included the assessment of pathological tumor response from on-therapy biopsies that were planned between the first and second infusion of pembrolizumab in patients with accessible disease and the determination of pathological responses to neoadjuvant pembrolizumab in surgical specimens from surgery after three or four infusions of neoadjuvant pembrolizumab (examples in Extended Data Figs. 3 and 4). Twenty-three of 28 patients had on-therapy biopsies (collected at a median of 21 days after the start of treatment, ranging from 19 to 31 days), but five biopsies did not contain tumor areas upon central pathology review. Among the 18 assessable on-therapy biopsies, 9 (50%) had no evidence of pathological features associated with a response to neoadjuvant immunotherapy, 2 (11%) showed some regions consistent with a pathological tumor response, and biopsies from 7 patients (39%) contained features consistent with a pathological response with no residual viable tumor.

In this multicenter US cooperative group clinical trial, surgical specimens submitted for central pathological assessment depended on local institutions and state regulations, with not all of them being

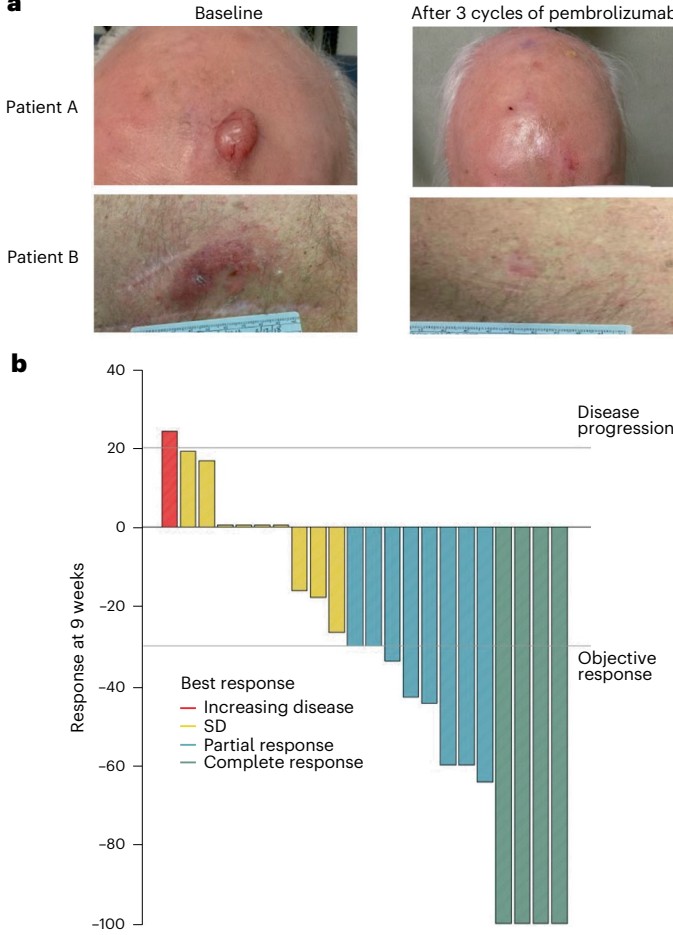

**a**

Baseline · After 3 cycles of pembrolizumab

Patient A

Patient B

**b**

**Fig. 3 | Clinical assessment of response to neoadjuvant pembrolizumab.**
**a**, Examples of clinical response noted after three cycles of pembrolizumab. A representative patient with a scalp lesion (top) and a representative patient with a back lesion (bottom) are shown. The pictures were obtained and published with patient written consent. **b**, Waterfall plot demonstrating the clinical response after 9 weeks of pembrolizumab (*n* = 22, excluding three patients who were not assessable for response). Clinical response was assessed radiographically or by clinical exam with photos. The vertical bars represent the percentage increase or decrease in the size of target lesions against baseline. Lines indicate the threshold for objective response (≥30% decrease) or disease progression (>20% increase); SD, stable disease.

**Table 2 | Number of patients with treatment-related adverse events**

| | Pembrolizumab (*N*=28) | |
|---|---|---|
| | **Any grade** | **Grade 3** |
| Any AE | 22 (79%) | 2 (7%) |
| Any SAE | 0 (0%) | 0 (0%) |
| AE/toxicity led to discontinuation of treatment | 1 (4%) | — |
| Immune-related AE[a] | 12 (43%) | 1 (4%) |
| SAE | 0 (0%) | 0 (0%) |
| AE led to death | 0 (0%) | 0 (0%) |

All adverse events reported were assessed as possibly, probably or definitely related to the study treatment. To be included in the safety analysis, patients must have received at least one dose of protocol therapy. Adverse event severity was scored using the National Cancer Institute Common Terminology Criteria for Adverse Events v4.0. There were no grade 4 or 5 (led to death) adverse events reported. [a]There was one incidence of grade 3 oral mucositis that was reported as a late effect and is unknown if deemed to be immune related. AE, adverse event; SAE, serious adverse event.

## Therapy-related adverse events

All 28 eligible patients who started on pembrolizumab were evaluated for safety (Fig. 1 and Extended Data Fig. 1). Twenty-two (79%) patients reported treatment-related adverse events of any grade (Table 2). Treatment-related adverse events with the highest frequency of all-grade events included fatigue (*n* = 12, 43%), maculopapular rash (*n* = 6, 21%) and diarrhea (*n* = 4, 14%; Supplementary Table 7). Two patients (7%) experienced grade 3 adverse events, one patient with mucositis and one patient with immune-mediated colitis, which led to discontinuation of treatment (Table 2). Both patients were able to undergo surgical resection.

## Survival outcomes

The median follow-up was 42 months (95% CI, 35–50 months). One patient refused surgery after neoadjuvant pembrolizumab, leaving 27 patients assessable for the analysis of RFS (Fig. 1b and Extended Data Fig. 1). The 3-year RFS rate among patients who underwent surgical resection was 74% (95% CI, 51–87%), and the median RFS has not been reached (Figs. 4 and 5a). The 3-year OS rate was 87% (95% CI, 65–96%), and the median OS has not been reached (Figs. 4 and 5b). Among the 28 patients assessable for survival, there were 6 relapse events, of which 3 were due to recurrent melanoma and 3 were death events of any cause. There have been four deaths, of which three were from unrelated causes and one was of unknown cause (Supplementary Table 2). The discrepancy between deaths in accounting for RFS and OS is because one patient had a relapse of melanoma before dying from another cause. The estimated 3-year melanoma-specific survival was 95% (95% CI, 80–100%), with the sole event being a patient who died of unknown cause(s) (Fig. 5c).

## Genomic analysis

Whole-exome sequencing analysis was a preplanned exploratory analysis. It was performed using 57 tumor samples (including 24 baseline, 27 on-therapy and 6 surgical samples from 26 patients) to assess tumor mutational burden (TMB) and canonical genetic drivers for desmoplastic melanoma (Fig. 6). Annotation from the centralized pathology review was used to remove whole-exome sequencing samples from tumor biopsies that did not have desmoplastic melanoma tumor content, such that ten patients were excluded from the analysis (Supplementary Table 8). The median TMB across these biopsies was 62.8 mutations per megabase (Mut/Mb; ranging from 0.43 to 160 Mut/Mb). This is an order of magnitude higher than in prior datasets of cutaneous melanoma, where the median TMB was 7.55 Mut/Mb (ref. 9). There was no significant difference in TMB comparing samples from patients with pCR and without pCR (Wilcoxon test, *P* = 0.79).

able to provide the complete surgical resection samples for central review. Three patients had no surgical specimens provided for central pathology review, and one additional patient did not undergo surgery (Extended Data Fig. 2). Among the 24 assessable surgical specimens, 5 (21%) had no evidence of pathological features associated with a response to neoadjuvant immunotherapy, 2 (8%) had a pathological PR (pPR), and 17 (71%) had a pCR. The pathological overall response rate (pCR and pPR) by central review was 79% (19 of 24; Supplementary Tables 5 and 6). The concordance between local and central pathological review was high, with 17 of the cases showing agreement between both reviews in calling a pCR, 5 concordant in agreeing to a no pCR, 1 case with no pCR by local review and considered a pPR by central review (PT0761) and 1 case considered to have a pCR by local review and a pPR by central review (PT0806). Of the 21 samples from patients with confirmed desmoplastic melanoma by central pathological review, 12 were annotated as pure desmoplastic histology and 9 as mixed desmoplastic histology. The pathological response rate by central review was 10 of 12 in those with pure histology and 7 of 9 in those with mixed histology. Notably, the two patients that had pPR or pPR with near pCR by central review showed mixed histology (Supplementary Table 5).

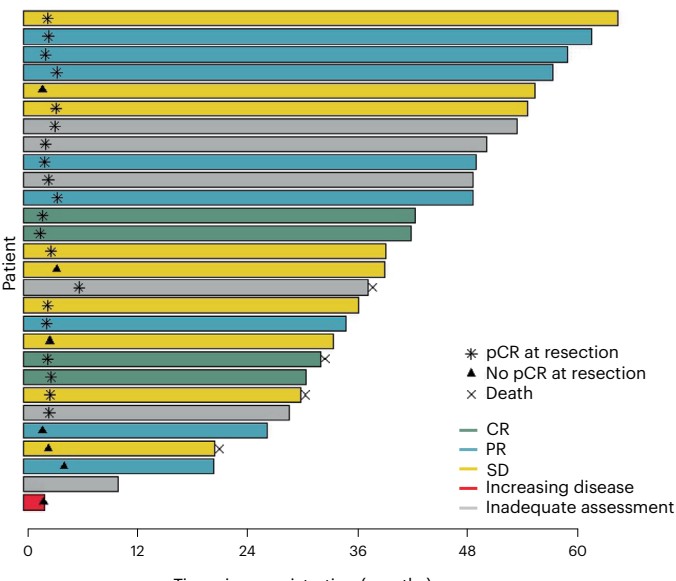

**Fig. 4 | Summary of individual patient outcomes.** Horizontal bars represent time from date of registration to date of death (indicated with an 'x') or date of last contact for the 28 patients in this cohort. The colors of the bars represent clinical response at 9 weeks. Three patients did not have measurable disease at baseline, and three patients were not assessable at 9 weeks; all six are labeled as 'Inadequate assessment' and shaded in gray. The asterisks and triangles are placed at the time of surgical resection and represent pathological response outcome. One patient did not undergo resection.

Tumor samples from 12 of 16 patients (75%) had *NF1* mutations; 10 of these were putative loss-of-function mutations (8 nonsense, 1 frameshift and 1 splice acceptor; Fig. 6), which is in agreement with prior reports in this cancer[3]. Other recurrently mutated genes included *ROS1* ($n = 12$, 75%), *TP53* ($n = 11$, 69%) and *CDKN2A* ($n = 7$, 44%). Six patients (37.5%) had biopsies with mutations in all four of these genes, and four additional patients had biopsies with mutations in three of these genes (25%). When the copy number profiles were evaluated across the cohort, whole-genome duplications were not prevalent, and there were no recurrent copy number changes (Extended Data Fig. 5). There was one outlier from this cohort (PT0761) in which the baseline biopsy showed low TMB (2.62 Mut/Mb) without a *BRAF*[V600E] mutation, and the on-therapy biopsy had a very low TMB (0.43 Mut/Mb) and a *BRAF*[V600E] mutation measured at 7.4% variant allele frequency. Following central pathology, it was agreed that this case most likely had two melanomas in one site, one a desmoplastic melanoma with mixed histology and the other potentially an acral melanoma with sheets of epithelioid cells with a radial growth phase and an in situ component (images in Extended Data Fig. 6). This patient had stable disease by RECIST (9 weeks following treatment) and pPR in the surgical resection with residual desmoplastic melanoma and epithelioid melanoma, as evaluated by local and central pathological review.

## Discussion

This prospective study of neoadjuvant anti-PD-1 therapy in patients with desmoplastic melanoma met its primary endpoint of a pCR rate of 71%, demonstrating that localized desmoplastic melanoma is highly responsive to systemic PD-1 blockade therapy. The high pCR rate supports the role of systemic treatment for localized desmoplastic melanoma before surgical intervention, after consideration of the potential toxicities induced by neoadjuvant anti-PD-1 therapy. pCRs were demonstrated when pembrolizumab was administered to patients either at initial diagnosis or at the time of locally recurrent disease. Neoadjuvant pembrolizumab was generally well tolerated, with the type of adverse events consistent with the known toxicity of this regimen[10]. Furthermore, none

of the patients became surgically unresectable. In prior clinical trials in patients with cutaneous melanoma, a single infusion of neoadjuvant pembrolizumab demonstrated a 19% pCR rate[11], three infusions of neoadjuvant pembrolizumab demonstrated a 21% pCR rate[6], and four infusions of nivolumab administered every 2 weeks demonstrated a pCR

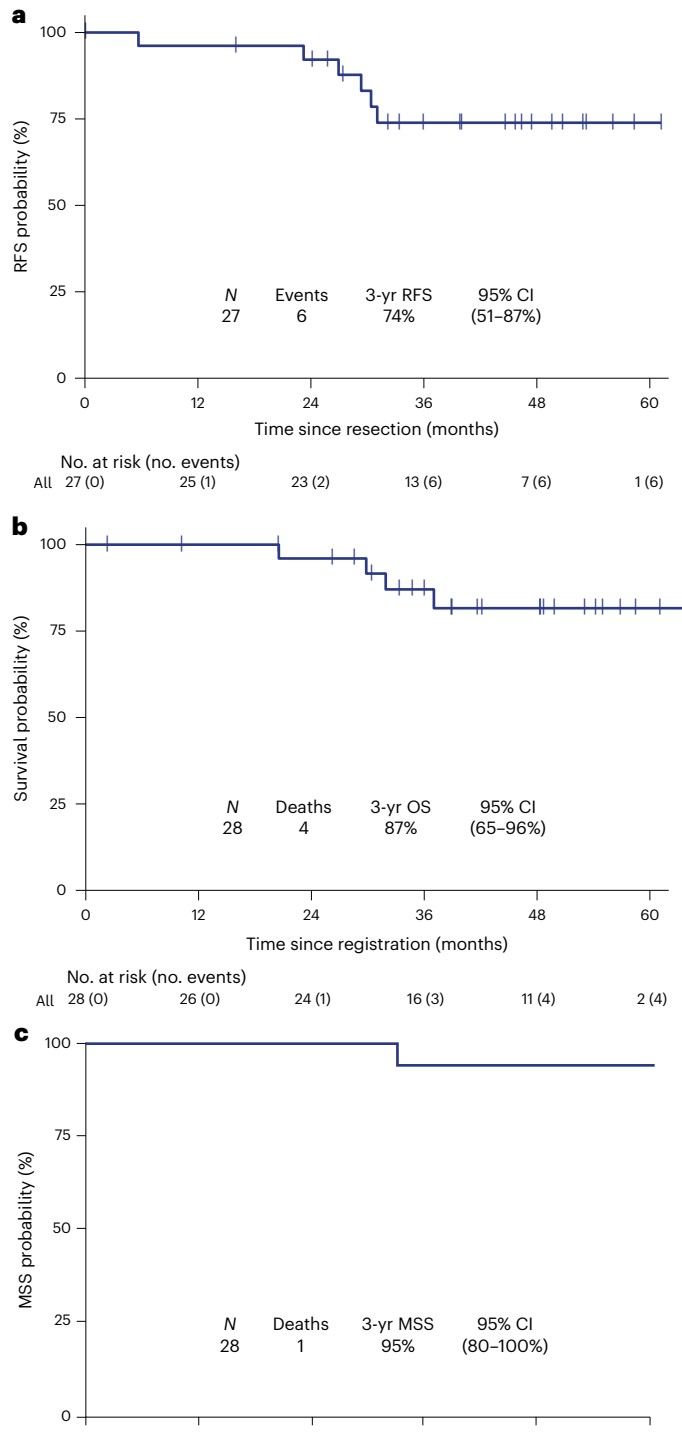

**Fig. 5 | Relapse-free and OS analyses. a**, Kaplan–Meier plot of RFS. The 3-year (yr) RFS estimate and 95% CI are reported. One patient did not undergo resection and is not included in the RFS analysis. **b**, Kaplan–Meier plot of OS. The 3-year OS estimate and 95% CI are reported. **c**, Melanoma-specific survival (MSS). The 3-year estimate and 95% CI are reported, with the single event being a patient who died from unknown causes.

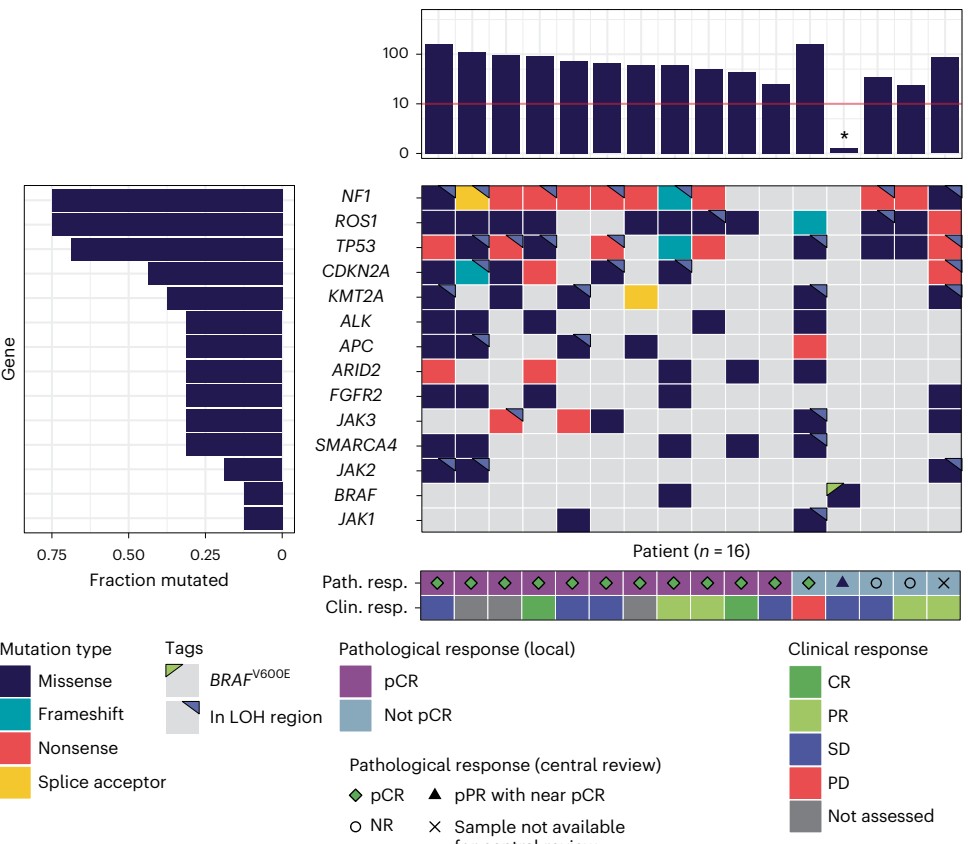

**Fig. 6 | TMB and driver oncogenic events.** Whole-exome sequencing data were available from 16 patients. TMB (in Mut/Mb, top) and genes mutated in at least 25% of cases are shown (bottom). Mutations are annotated by their putative impact (fill color) and whether they are present in chromosomal regions with loss-of-heterozygosity (LOH). Patients are further annotated based on their pathological response (Path. resp.) determined by local (fill color) or central (shapes) review of surgical sections. Clinical response (Clin. resp.) by imaging at 9 weeks is shown for patients with CR, PR, stable disease or progressive disease (PD); NR, no response.

rate of 25% (ref. 12). The pCR rate in our series of patients with desmoplastic melanoma was also higher than that noted with combination nivolumab and ipilimumab in patients with cutaneous melanoma, with pCR rates ranging from 30% to 57% (refs. 7,12,13) and the nivolumab and relatlimab combination demonstrating a pCR rate of 59% (ref. 14). The benefit of neoadjuvant pembrolizumab was generally consistent across all subgroups, including age, sex, performance status, lactate dehydrogenase level and disease status and in patients with pure and mixed desmoplastic melanoma histological subtypes.

A limitation of this work is having pCR as the primary endpoint. The decision to select pCR by local pathology review as the primary endpoint for this clinical trial was based on an endpoint assessable at every study site of this US cooperative group clinical trial (it becomes a yes or no answer to the question of whether there is evidence of melanoma cells in the resected surgical specimen). We planned a central pathological review as an exploratory endpoint, with the knowledge that the local and state regulations in the SWOG sites across the US vary widely, and many sites have mandates to keep the surgical samples for any future need to rereview them. The S1512 clinical trial was activated in 2015, which was before the criteria for assessment of pathological responses to neoadjuvant ICB therapy had been developed by the International Neoadjuvant Melanoma Consortium[15,16]. Therefore, we had to develop our own criteria for this protocol adapted to the particular histology of desmoplastic melanoma. For the central review, we decided to assess both pCR and pPR, with the consensus to analyze the suspected percent antitumor immune response by pathological response criteria visualizing areas of fibrosis or necrosis where the cancer cells used to be present. Such an analysis would not be standardized at each local

pathology site and would not be reliable by local pathological analysis. Regardless, there was high concordance between the local and central reviews of surgical specimens. The central pathological review also allowed for the assessment of changes in time of an ongoing antitumor response, or lack of, induced by neoadjuvant anti-PD-1 therapy. We had collected optional on-therapy research biopsies in consenting patients, and these were analyzed centrally, comparing them with the baseline biopsies and the specimens submitted from the surgical excision. The serial samples provided interesting information about the time-course evolution of the antitumor immune response induced by pembrolizumab; only half of the earlier on-therapy biopsies had pathological evidence of response to immunotherapy, whereas the later surgical specimens had 79% evidence of pathological response to neoadjuvant anti-PD-1 immunotherapy by central pathological review.

We reasoned that the demonstration of a high rate of pCR with neoadjuvant anti-PD-1 in patients with localized desmoplastic melanoma may help in the future testing of scaled-down plans for surgeries, with less frequent performance of highly morbid surgical interventions in highly visible areas. The SWOG S1801 and the NADINA clinical trials support the overall idea that neoadjuvant ICB therapy improves clinical outcomes[6,17] and demonstrated that pathology-based treatment de-escalation is feasible and safe[17]. Dogma would dictate that patients not having a pCR would have a high risk of recurrence and death from melanoma, but current data do not clearly demonstrate this in patients with desmoplastic melanoma. Of note, one patient in this series refused surgery as they had a complete clinical response and has not had a relapse, which is consistent with occasional patients who had clinical responses to neoadjuvant immunotherapy and did not have

surgery in other series[6,18]. Although allowed per protocol, none of the patients received adjuvant systemic therapy after surgery in S1512, which is an approved standard of care treatment for patients with surgically resected melanoma[19]. This may reflect that the investigators and patients had the perception of not needing further therapy after the neoadjuvant pembrolizumab and surgery.

Melanoma subtypes vary in their response to immunotherapies, and their relationship to the carcinogenic effects of UV light-induced mutations may have an important impact on response rates. Particularly, mucosal, acral and uveal melanomas have little to no evidence of UV-mediated DNA damage and have lower response rates to ICB therapies than cutaneous melanomas, whereas desmoplastic melanomas have higher evidence of UV-mediated DNA damage with a high TMB, which is associated with high response rates to anti-PD-1 immunotherapy[2]. The genomic profiles of the biopsies from our patient cohort support a high degree of genomic homogeneity, with recurrent mutations in tumor suppressor genes *NF1*, *TP53* and *CDKN2A* and high TMB[3]. Previous neoadjuvant studies in melanoma enrolled patients with cutaneous melanomas independent of the subtype, with potential to dilute out the response of those subtypes more likely to respond to immunotherapies, as we are presenting with desmoplastic melanoma.

In conclusion, cohort A of S1512 demonstrates a high rate of pCR in patients with resectable desmoplastic melanoma receiving neoadjuvant infusions of pembrolizumab, with generally manageable toxicities and excellent long-term outcomes, with none of the patients in this series having been reported to die from melanoma or side effects of the treatment. These data provide evidence that three doses of neoadjuvant single-agent anti-PD-1 therapy could be considered before surgery for patients with resectable desmoplastic melanoma.

## Methods

### Trial design and treatments

This was a single-arm, phase 2 study within the two-cohort SWOG S1512 clinical trial. Pembrolizumab at 200 mg was given intravenously every 3 weeks for a total of three doses before surgery, with an optional on-therapy biopsy at 3 to 5 weeks from the start of therapy. Tumor measurements were performed before initiation of therapy and at 9 weeks, before surgical resection. If the patient's disease was deemed to be unresectable, the patient was allowed another cycle of pembrolizumab followed by remeasurement to distinguish pseudoprogression as opposed to true progression. After discontinuation of treatment, if the patient's disease was potentially resectable, the patient was to proceed with surgical resection. The approach used for surgical resection was at the discretion of the surgeon performing the procedure using best clinical judgment and institutional guidelines. Repeat surgeries were allowed. The intent was for all sites of disease to be completely resected. Sentinel lymph node biopsy was not required for clinically node-negative primary cases but could be performed at the surgeon's discretion. If sentinel lymph nodes were positive, nodal basin observation or completion lymphadenectomy was performed per institutional guidelines. Resected margins were defined as R0 (negative margins), R1 (histologically positive margins with no grossly visible tumor) and R2 (histologically positive margins with partial resection with grossly visible tumor left behind). Tissue was obtained for correlative analysis before the first pembrolizumab treatment and at the time of surgical resection.

### Trial oversight

Approval for the trial was obtained through the Cancer Therapy Evaluation Program Central Institutional Review Board. The trial was conducted in accordance with the principles of the Declaration of Helsinki and Good Clinical Practice guidelines, and the study protocol was approved by the relevant ethics bodies at each participating site. Patients provided written informed consent before being screened for enrollment. The authors affirm that human research participants provided informed consent for publication of the images in Fig. 3a. The clinical trial was registered in Clinicaltrials.gov under number NCT02775851.

### Participants

Patients were eligible if they were at least 18 years of age with histologically or cytologically confirmed diagnosis of desmoplastic melanoma deemed surgically resectable by the judgment of the surgeon. Patients could have measurable disease according to RECIST version 1.1 or nonmeasurable disease confirmed with fine needle aspirate. Eligible patients were required to have adequate bone marrow function, adequate liver function and a performance status of ≤2. Prior surgery was allowed. Both a primary desmoplastic melanoma and locally recurrent disease were eligible. Main exclusion criteria included prior systemic treatment for melanoma, active autoimmune disease requiring systemic treatment within 2 years of study entry or live vaccines within 42 days of registration. Full information on imaging requirements and eligibility criteria is included in the clinical trial protocol in the Supplementary Information.

### Outcomes

The primary endpoint was the rate of pCR in the surgical specimen, defined as no evidence of viable cancer cells on complete pathological evaluation of the surgically resected specimen per institutional standard of care using local pathology assessment. Secondary endpoints included clinical response rate, OS, RFS among resected patients and toxicities. Disease assessments, radiographically or by clinical exam documented by photographs with a ruler, occurred at baseline and at 9 weeks after the start of pembrolizumab. RECIST version 1.1 criteria were used for clinical response assessment among all patients with measurable disease at baseline. OS was measured from registration to the date of death from any cause, with patients last known to be alive censored at the date of last contact. RFS was measured from the date of resection to the date of first documentation of progression, symptomatic deterioration or death due to any cause, with patients last known to be alive and without report of progression censored at the date of last contact. Adverse events were assessed by the investigator using the National Cancer Institute Common Terminology Criteria for adverse Events, version 4.03. Clinical trial enrollment was open from 26 July 2017 to 20 May 2021.

### Central pathological evaluation

The pathological diagnosis of desmoplastic melanoma was assessed by histologic review by two pathologists (D.A.W. and J.A.P.), following standardized criteria established before the independent reviews. Desmoplastic melanoma was defined by previously published criteria of the presence of spindle tumor cells, mostly with elongated hyperchromatic nuclei and mild or moderate cytologic atypia, with abundant stromal collagen (desmoplasia)[20–22]. 'Pure' desmoplastic melanoma was defined as >90% of the tumor containing a low density of tumor cells as described above, imparting a scar-like, low-power (×20) histologic pattern. 'Mixed' desmoplastic melanoma contained significant (>10%) areas of morphologically distinct tumor with a higher cell density and less stromal collagen and/or the presence of epithelioid tumor cells[22]. Desmoplastic melanoma cells would be S100+ and/or SOX10+ and negative for HMB45, gp100 and tyrosinase and often negative for Melan-A. Pathological review for confirmation of diagnosis evaluated the presence or absence of pure or mixed desmoplastic features, perineural invasion and immunophenotype staining.

### Sample collection, nucleic acid extraction and whole-exome sequencing

Biopsies and whole-blood samples individually procured at each study site were sent to the University of California, Los Angeles, for centralized processing for pathological staining, nucleic acid extraction and

whole-exome sequencing as previously described[8]. Each sample was analyzed for melanoma genetic subtypes (*BRAF* activating mutations, *NRAS* hotspot mutations and *NF1* mutant or loss-of-function mutations). Nonsilent TMB was quantified by normalizing the number of nonsilent variants with at least 5% variant allele frequency by the size of the genome (in megabases) with at least 50× depth of coverage. If there were multiple biopsies collected for a given patient, samples were evaluated for both high tumor cellularity (by manual review of immunohistochemistry and S100-stained slides) and adequate sequencing coverage.

### Statistical analysis

The full details of the design are provided in Section 11 of the protocol document in the Supplementary Information. The primary endpoint was the rate of pCR. The sample size ($n = 25$) was based on a single-stage design with 90% power to rule out a pCR rate of 5% (null pCR rate) at the 3.4% level, if the true pCR rate was 25% (alternative pCR rate). The observation of 4 of 25 cases with pCR would be considered evidence that the treatment warrants further study, provided other factors such as toxicity and OS also appear favorable. Enrollment of 30 patients was estimated to allow for at least 25 evaluable patients. All patients who received a single dose of study treatment and met eligibility criteria were considered evaluable for the primary endpoint. Patients who were unable to have all sites of disease completely resected and patients who did not undergo surgery on protocol were counted as nonresponders. A one-sided exact binomial test using the method of Clopper and Pearson was used to test the pCR rate against the null hypothesis of 5% (ref. 23). Overall response rate was assessed among patients with measurable disease at baseline who received a single dose of study treatment. Patients with inadequate follow-up assessments were analyzed as nonresponders per RECIST version 1.1 criteria. Binary proportions are summarized along with 95% CIs. The method of Kaplan–Meier was used to estimate the distributions of RFS and OS, and the log–log method was used to estimate the corresponding CIs for survival at 3 years. Melanoma-specific survival was calculated as 1 − CIF, where CIF is the cumulative incidence of melanoma-specific deaths estimated nonparametrically using the method of Nelson–Aalen. Death from other causes was treated as a competing risk. Melanoma-specific survival CIs for survival at 3 years were calculated using the method recommended by Pintilie[24]. Median follow-up was estimated using the reverse Kaplan–Meier approach. All analyses were performed using SAS 9.4 (SAS Institute) and R v4.3.1 (The R Foundation for Statistical Computing).

### Reporting summary

Further information on research design is available in the Nature Portfolio Reporting Summary linked to this article.

## Data availability

SWOG makes all research data available externally to investigators and pharmaceutical companies, in accordance with the policies of the NIH and NCI. All data to reproduce the analyses presented in this article are available upon request from SWOG in accordance with SWOG's data sharing policy and process: https://www.swog.org/sites/default/files/docs/2019-12/Policy43_0.pdf. The protocol and informed consent are found in the Supplementary Information. Whole-exome sequencing data are available with authorized access upon registration and approval via dbGaP under accession number phs004123.v1.p1. The remaining data are available within the article and Supplementary Information. Source data are provided with this paper.

## Code availability

All code to reproduce the analyses presented in this article is available upon request from SWOG in accordance with SWOG's data sharing policy and process (see Data availability). The code for genomic analysis is available at https://github.com/kcampbel/s1512_public.

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

## Acknowledgements

We acknowledge the participation of the patients and their caregivers, the support of the patient advocates S. Guild and V. Guild (deceased) and the support of Merck & Co in providing investigational agents for the study. The clinical trial was funded by SWOG National Institutes of Health/National Cancer Institute (NIH/NCI) grants LS1616_R01LDRGAPP01, U10CA180888, U10CA180819, U10CA180821 and U10CA18068. A.R. was funded by NIH/NCI grants P01 CA244118 and R35 CA197633, the Agilent Thought Leader Award, the Parker Institute for Cancer Immunotherapy, the Ressler Family Fund and support from M. Tanner and M. Grimaldi, K. and D. Schultz, T. and D. Jones and M. J. and R. Rumer. K.M.C. was supported by the Cancer Research Institute Postdoctoral Fellowship Program, the V Foundation Gil Nickel Melanoma Research Fellowship and the Parker Institute for Cancer Immunotherapy and V Foundation Bridge Fellowship. N.N.A.D. was funded by The Alan Ghitis Fellowship Award for Melanoma Research. The NCI approved the study design and through its grants to SWOG supported the conduct of the study and collection, management, analysis and interpretation of the data and, through the melanoma committee of SWOG, supported the preparation, review and approval of the paper and the decision to submit the paper for publication. Merck approved the design but had no role in the conduct of the study; collection, management, analysis and interpretation of the data; preparation, review and approval of the paper or decision to submit the paper for publication. The content is solely the responsibility of the authors and does not necessarily represent the official views of the NIH.

## Author contributions

K.L.K., S.H.-L., W.E.C., S.P.P., E.S., M.C.W. and A.R. designed, initiated and oversaw the conduct of the clinical trial. K.L.K., Z.E., S.H.-L., W.E.C., G.K.I., A.I., J.H., A.S.B., B.C., N.I.K., J. Markowitz, M.M., C.M.C., T.B., K.N., K.F.G., V.K.S. and A.R. enrolled, treated and cared for patients on the clinical study protocol. K.L.K., S.L.B., J. Moon, M.C.W. and A.R. conducted clinical data analyses. K.M.C., E.M., I.B.-C., C.R.G., I.P.G., A.V.-C., J.M.C. and N.N.A.D. processed, banked and analyzed biopsies. D.A.W. and J.A.P. performed central pathological analysis of biopsies. K.M.C., D.A.W., J.A.P., E.M., C.R.G., N.N.A.D. and A.R. interpreted biopsy analyses. K.L.K., S.L.B., K.M.C., M.C.W. and A.R. wrote the paper. All authors proofread and approved the final paper.

## Competing interests

K.L.K. reports institutional research support from Bristol Myers Squibb and trial support from GlaxoSmithKline, Immunocore, Varian Medical Systems and Merck. Z.E. is on advisory boards for Regeneron, Pfizer, Replimune, Incyte, Natera and SunPharma and receives research funding from Pfizer and Boehringer Ingelheim. S.H.-L. is a scientific advisor/consultant for Amgen, Ascendis, Astellas, BMS, Genmab, Endeavor, Immunocore, Merck, Nektar, Neon Therapeutics, Novartis, Regeneron, Replimune, Vaccinex and Xencor and contracted research through affiliated institutions with Astellas, Aulos Bio, BioAtla, BMS, Boehringer Ingelheim, Checkmate, Dragonfly, Erasca, F Star, Genentech, Immunocore, Iovance, Kite Pharma, Lyell, Merck, Nektar, Neon Therapeutics, OncoC4, Pfizer, Plexxikon, Vaccinex, Vedanta and Xencor. K.M.C. reports being a shareholder in Geneoscopy and Georgiamune and has received consulting fees from Geneoscopy, PACT Pharma, Tango Therapeutics, Flagship Labs 81, the Rare Cancer Research Foundation, the Jaime Leandro Foundation, Noetik, AME Therapeutics and Georgiamune. D.A.W. reports clinical trial support from Orlucent and Blueprint Medicines. G.K.I. reports institutional research grants/contracts from Pfizer, Regneron, Replmune, Bicara, Merck, Georgiamune, Obsidian, Immunocore, Iovance and Xencor; serves on advisory boards for Pfizer, Regeneron, Replimune and Obsidian and is a consultant for Pfizer. A.I. reports research funding from Merck. J.H. reports institutional research grants/contracts from Merck, BMS, Iovance, Lyell, Natera, Skyline and Philogen. A.S.B. is an advisory board member for Deciphera and received research funding from Merck. B.C. serves on advisory boards for Atreca, Regeneron and Treeline Biosciences; is on the data monitoring committee for Servier and SpringWorks Therapeutics and receives clinical trial support from Bristol Myers Squibb, Macrogenics, Infinity Pharmaceuticals, Advenchen Laboratories, Xencor, Compugen, Iovance, RAPT Therapeutics, IDEAYA Biosciences, Ascentage, Atreca, Replimune, InstilBio, Adagene, TriSalus Life Sciences, Kinnate, PTC Therapeutics, Xilio Therapeutics, Kezar Life Sciences, Immunocore, AskGene Pharma, Krystal Bio, Nested Therapeutics, Pierre Fabre, Georgiamune and Immatics. N.I.K. serves on advisory boards for Regeneron, Merck, Replimune, Immunocore, Iovance Biotherapeutics, Novartis, IO Biotech, MyCareGorithm and HUYABIO International; receives travel support from Castle Biosciences and Regeneron; is on the data safety monitoring board for Incyte and AstraZeneca; serves on the scientific advisory board for T-Knife Therapeutics; is on the study steering committees for BMS, Nektar, Regeneron and Replimune; owns common stock in Bellicum Pharmaceuticals and Amarin and receives research funding (to institute) from BMS, Merck, Regeneron, Replimune, GSK, Celgene, Novartis, IDEAYA Biosciences, Modulation Therapeutics and HUYABIO International. J. Markowitz receives research support from Morphogenesis (now TuHURA Biosciences) and Merck. M.M. serves on advisory boards for Merck and Regeneron. T.B. receives institutional research support from Replimune, Genentech, Amgen, Natera and Iovance. S.P.P. receives honoraria from advisory boards, steering committees, data safety monitoring boards or consulting from Bristol Myers Squibb, Cardinal Health, Castle Biosciences, Ideaya, Immatics, IO Biotech, MSD, Novartis, Obsidian, OncoSec, Pfizer, Replimune, Scancell and TriSalus Life Sciences. K.F.G. was employed by Merck from November 2021 to August 2023. V.K.S. is a consultant for Bristol Myers Squibb, Genesis Drug Discovery and Development, Merck, Mural Oncology and Novartis and receives research funding from Neogene Therapeutics, Skyline and Turnstone. E.S. is a consultant for DE Shaw Research and serves on the advisory board for Mallinckrodt Pharmaceuticals. A.R. has received honoraria from consulting for Amgen, Bristol Myers Squibb, Merck, Novartis and Roche-Genentech; is or has been a member of the scientific advisory board; holds stock in Appia, Apricity, Arcus, Compugen, CytomX, ImaginAb, ImmPact, Inspirna, Kite-Gilead, Larkspur, Lyell, Lutris, MapKure, Merus, Synthekine and Tango and has received research funding from Agilent and Bristol Myers Squibb through Stand Up to Cancer (SU2C) and patent royalties from Arsenal Bio. S.L.B., W.E.C., J.A.P., C.M.C., E.M., I.B.-C., C.R.G., K.N., I.P.G., A.V.-C., J.M.C., N.N.A.D., J. Moon and M.C.W. report no competing interests.

## Additional information

**Extended data** is available for this paper at https://doi.org/10.1038/s43018-025-01113-y.

**Correspondence and requests for materials** should be addressed to Kari L. Kendra or Antoni Ribas.

[1]Ohio State University Wexner Medical Center, Columbus, OH, USA. [2]SWOG Statistics and Data Management Center, Seattle, WA, USA. [3]Fred Hutchinson Cancer Center, Seattle, WA, USA. [4]H. Lee Moffitt Cancer Center and Research Institute, Tampa, FL, USA. [5]Huntsman Cancer Institute, University of Utah, Salt Lake City, UT, USA. [6]Jonsson Comprehensive Cancer Center, University of California, Los Angeles, CA, USA. [7]Norris Comprehensive Cancer Center, University of Southern California, Los Angeles, CA, USA. [8]University of Oklahoma Stephenson Cancer Center, Oklahoma City, OK, USA. [9]Intermountain Medical Center, Murray, UT, USA. [10]Western States Cancer Research, Jefferson Healthcare, Seattle, WA, USA. [11]University of Colorado Cancer Center, Aurora, CO, USA. [12]Cancer Therapy Evaluation Program, Division of Cancer Treatment and Diagnosis, National Cancer Institute, Bethesda, MD, USA. [13]Present address: University of Texas, MD Anderson Cancer Center, Houston, USA. [14]Present address: RUSH MD Anderson Cancer Center, Chicago, USA. [15]Present address: Robert W. Franz Cancer Research Center, Earle A. Chiles Research Institute, Providence Cancer Center, Providence Portland Medical Center, Portland, OR, USA. [16]Present address: Dana Farber Cancer Institute, Boston, MA, USA. [17]These authors jointly supervised this work: Kari L. Kendra, Antoni Ribas. ✉e-mail: kari.kendra@osumc.edu; aribas@mednet.ucla.edu

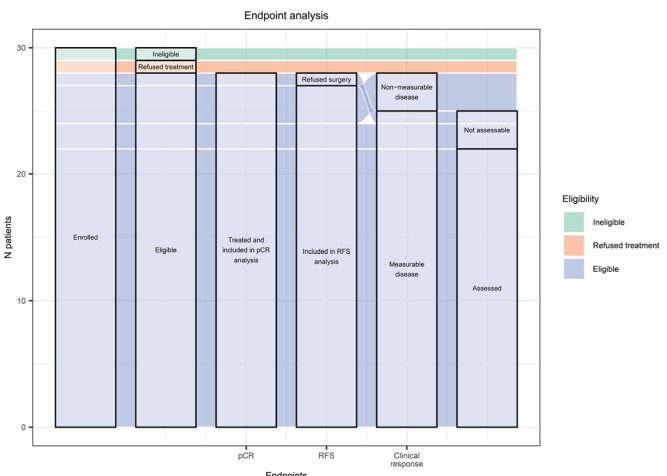

**Extended Data Fig. 1 | Patient inclusion in endpoint analyses.** This alluvial diagram depicts the inclusion or exclusion of patients in the analysis of each study endpoint (x-axis). Patients that were treated with pembrolizumab were included in the analysis regarding that pathological complete response (pCR) rates. Patients that were treated with pembrolizumab and underwent surgery were included in the analysis for the relapse-free survival (RFS) rate. Patients that were treated with pembrolizumab and had measurable disease were included in the analysis for clinical response, even if they were not assessed by imaging at 9 weeks.

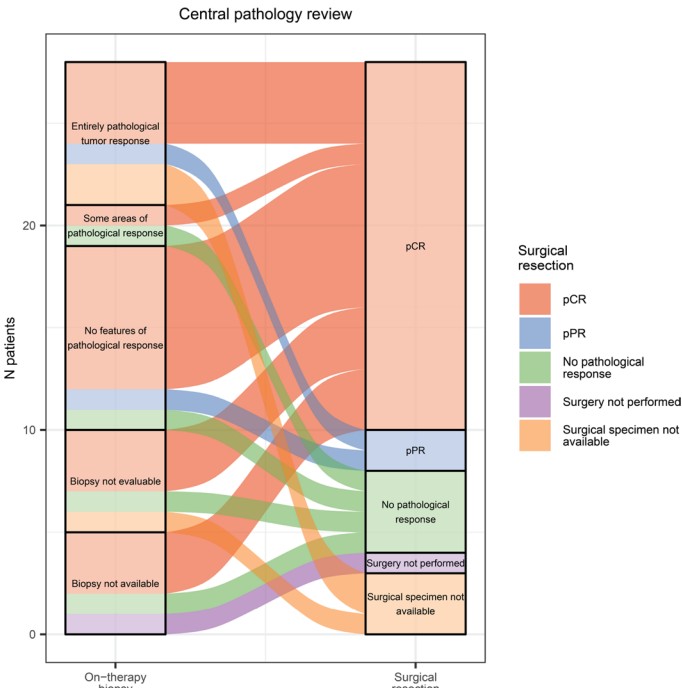

Central pathology review

**Extended Data Fig. 2 | Summary of central pathological review.** This alluvial plot demonstrates the annotation of on-therapy biopsies and surgical resection specimens (x-axis) based upon a centralized review. Alluvia are colored based upon the pathologic annotation of the surgical resection. Out of 28 patients and treated in this study, we were able to complete a centralized review of samples from 24 patients. The figure highlights that there was a progressive improvement in the evidence of pathological response from the earlier on-therapy biopsies to the later surgical resection specimens.

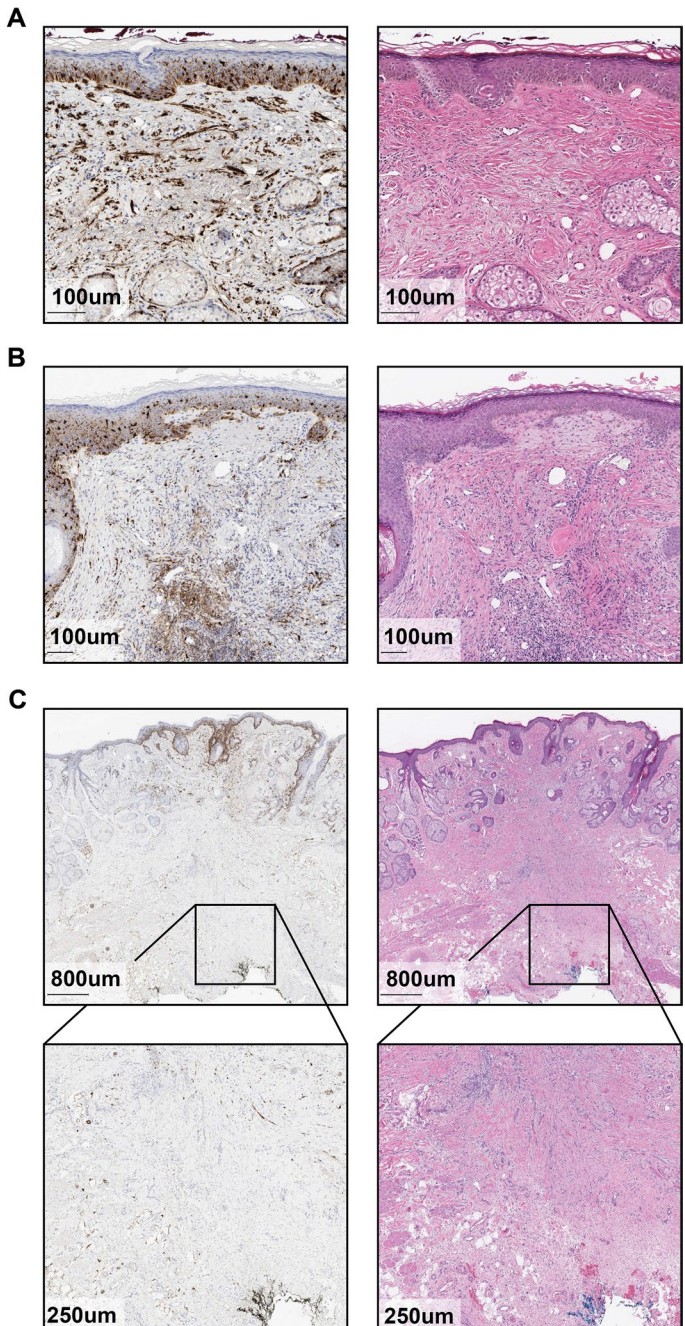

**Extended Data Fig. 3 | Example of central pathological review of samples from a patient with evidence of pathological response (PT0776).** The central pathology review confirmed (**A**) desmoplastic melanoma in the diagnostic biopsy, based upon S100-stained (left) and H&E (right) images, (**B**) treatment effect and features of pathological response in the on-therapy biopsy, and (**C**) pathological complete response (pCR) and absence of viable tumor in the surgical sample.

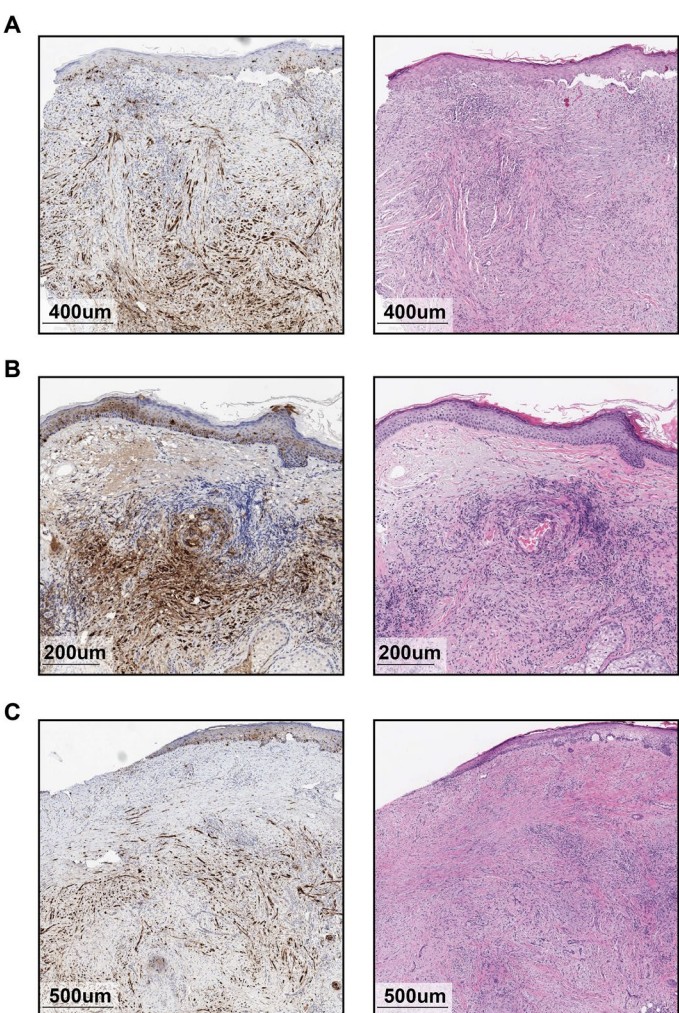

**Extended Data Fig. 4 | Example of central pathological review of samples from a patient with no evidence of pathological response (PT0665).** The central pathology review confirmed (**A**) desmoplastic melanoma in the diagnostic biopsy, based upon S100-stained (left) and H&E (right) images, (**B**) desmoplastic melanoma in the on-therapy biopsy, and (**C**) no pathological tumor response in the surgical sample.

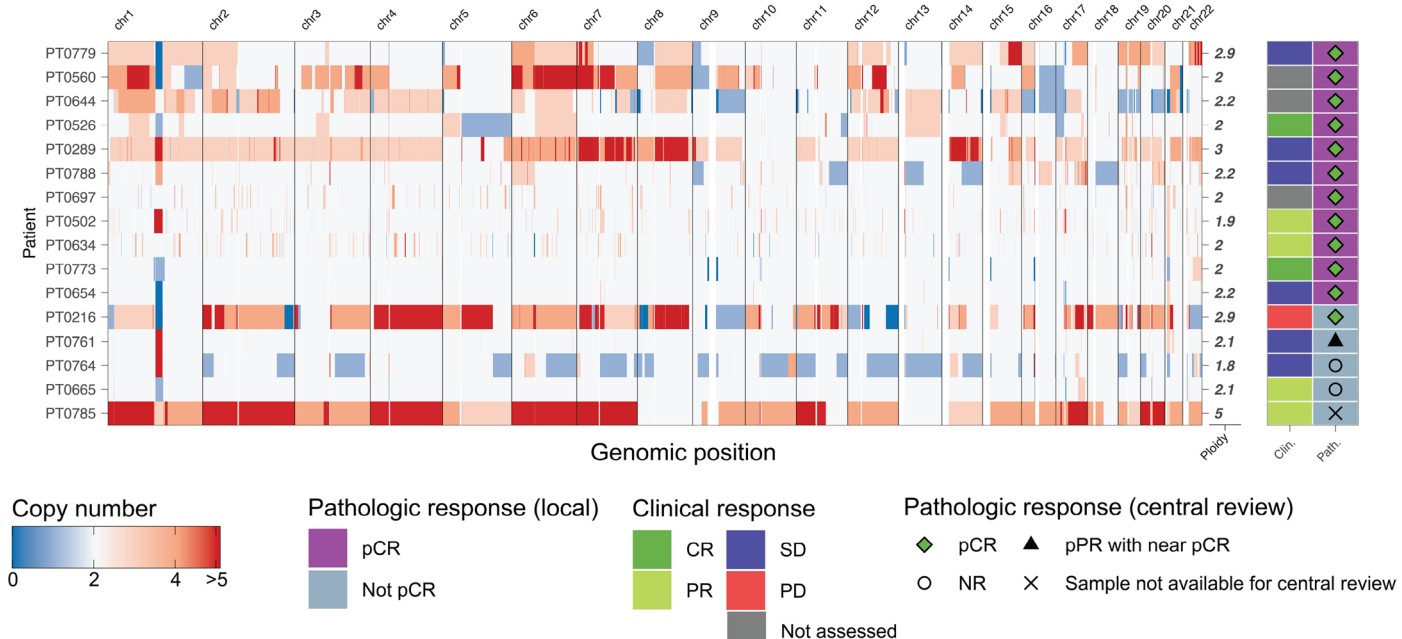

**Extended Data Fig. 5 | Copy number segments** (organized by chromosome along the x-axis), called from whole exome sequencing (WES) data by Sequenza, are shown based upon the total copy number (fill color). Profiles are organized in the same order as the genomic profiles shown in Fig. 4, with patients (rows) annotated by clinical responses and pathological responses (based upon both local [fill color] and centralized [point annotations]) on the right. Each sample is further annotated with its overall ploidy across the whole genome.

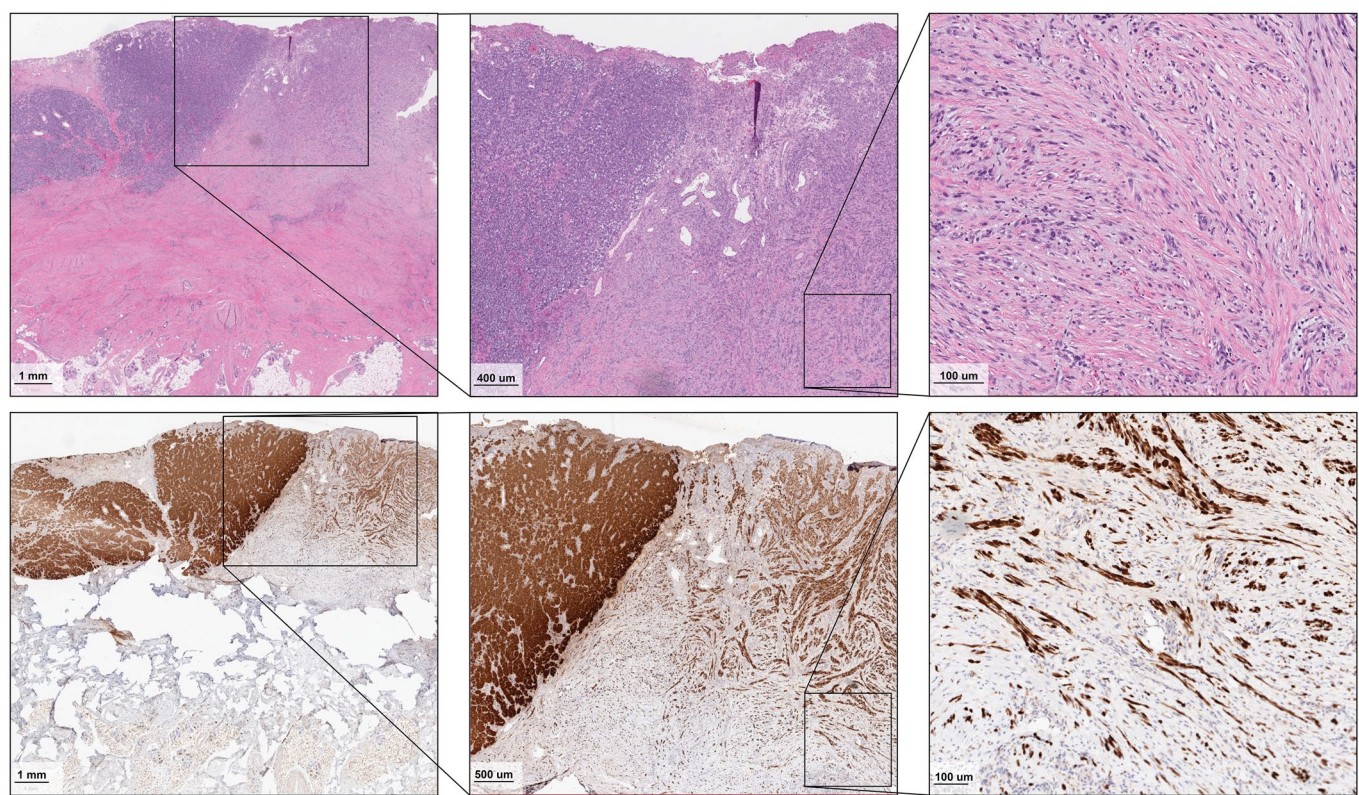

**Extended Data Fig. 6 | Diagnostic biopsy analysis of PT0761.** This case has evidence of two different histologies of melanoma, which may be co-primary melanomas at the same site. **A**) Low power image of the specimen by H&E and S100 IHC, showing a dense sheet of S100 positive epithelioid melanoma cells in the left side, with evidence of radial growth phase and an *in situ* component (higher power image in **B**), and an area in the right of the specimen with desmoplastic melanoma of pure histology (high power image in **C**).

# Reporting Summary

## Statistics

For all statistical analyses, confirm that the following items are present in the figure legend, table legend, main text, or Methods section.

| n/a | Confirmed | |
|---|---|---|
| ☐ | ☒ | The exact sample size ($n$) for each experimental group/condition, given as a discrete number and unit of measurement |
| ☐ | ☒ | A statement on whether measurements were taken from distinct samples or whether the same sample was measured repeatedly |
| ☐ | ☒ | The statistical test(s) used AND whether they are one- or two-sided<br>*Only common tests should be described solely by name; describe more complex techniques in the Methods section.* |
| ☒ | ☐ | A description of all covariates tested |
| ☐ | ☒ | A description of any assumptions or corrections, such as tests of normality and adjustment for multiple comparisons |
| ☐ | ☒ | A full description of the statistical parameters including central tendency (e.g. means) or other basic estimates (e.g. regression coefficient) AND variation (e.g. standard deviation) or associated estimates of uncertainty (e.g. confidence intervals) |
| ☐ | ☒ | For null hypothesis testing, the test statistic (e.g. $F$, $t$, $r$) with confidence intervals, effect sizes, degrees of freedom and $P$ value noted<br>*Give P values as exact values whenever suitable.* |
| ☒ | ☐ | For Bayesian analysis, information on the choice of priors and Markov chain Monte Carlo settings |
| ☒ | ☐ | For hierarchical and complex designs, identification of the appropriate level for tests and full reporting of outcomes |
| ☐ | ☒ | Estimates of effect sizes (e.g. Cohen's $d$, Pearson's $r$), indicating how they were calculated |

*Our web collection on statistics for biologists contains articles on many of the points above.*

## Software and code

Policy information about availability of computer code

| | |
|---|---|
| Data collection | Clinical data from sites reported through iMedidate Rave; transferred to the SWOG SQL database; exported for analysis in R (v4.2.0). |
| Data analysis | All analyses were performed using SAS 9.4 (SAS Institute Inc.) and R v4.3.1 (The R Foundation for Statistical Computing). |

For manuscripts utilizing custom algorithms or software that are central to the research but not yet described in published literature, software must be made available to editors and reviewers. We strongly encourage code deposition in a community repository (e.g. GitHub). See the Nature Portfolio guidelines for submitting code & software for further information.

## Data

Policy information about availability of data

All manuscripts must include a data availability statement. This statement should provide the following information, where applicable:
- Accession codes, unique identifiers, or web links for publicly available datasets
- A description of any restrictions on data availability
- For clinical datasets or third party data, please ensure that the statement adheres to our policy

SWOG makes all research data available externally to investigators and pharmaceutical companies, in accordance with the policies of the National Institutes of

## Human research participants

Policy information about studies involving human research participants and Sex and Gender in Research.

| | |
|---|---|
| Reporting on sex and gender | Patients were screened and enrolled on this study irrespective of their sex/gender. Any data regarding a patient's sex and gender was collected by the clinical trial groups at each site. Sex- or gender-based subgroup analysis are reported in Table 1 and Extended Data Table 1. |
| Population characteristics | Provided in Extended Data Table 1. |
| Recruitment | Patients were recruited across 10 clinical investigational sites. Given that this was an NCI-funded US cooperative group trial, the sites were limited to the United States of America. There were no biases introduced and patients were screened on a first-come first-serve basis based on meeting the protocol inclusion-exclusion criteria. No protocol waivers were allowed on this study. |
| Ethics oversight | The trial was conducted in accordance with the principles of the Declaration of Helsinki. The trial protocol and statistical analysis plan were designed in a collaboration between the SWOG and CTEP investigators. The protocol was approved by the Cancer Therapy Evaluation Program (CTEP) Central Institutional Review Board (CIRB)and institutional review boards from each of the 10 clinical sites enrolling patients to cohort A of S1512. |

Note that full information on the approval of the study protocol must also be provided in the manuscript.

# Field-specific reporting

Please select the one below that is the best fit for your research. If you are not sure, read the appropriate sections before making your selection.

☒ Life sciences ☐ Behavioural & social sciences ☐ Ecological, evolutionary & environmental sciences

For a reference copy of the document with all sections, see nature.com/documents/nr-reporting-summary-flat.pdf

# Life sciences study design

All studies must disclose on these points even when the disclosure is negative.

| | |
|---|---|
| Sample size | The primary endpoint was the rate of pathologic complete response (pCR). The sample size (n=25) was based on a single stage design with 90% power to rule out a pCR rate of 5% at the 3.4% level, if true pCR rate was 25%. The observation of four out of 25 cases with pCR would be considered evidence that the treatment warrants further study, provided other factors such as toxicity and overall survival also appear favorable. Enrollment of 30 patients was estimated to allow for at least 25 evaluable patients. All patients who receive a single dose of study treatment and met eligibility criteria were considered evaluable for the primary endpoint. |
| Data exclusions | 28 of 30 enrolled patients were included in the analysis. One patient refused protocol therapy and withdrew consent, and one patient was deemed ineligible after a review of the pathology report indicated their disease was not consistent with desmoplastic melanoma. |
| Replication | N/A - phase 2 clinical trial |
| Randomization | Not randomized. |
| Blinding | Study are was no blinded. No placebo was given and so no blinding was possible, per standard with many oncology studies. |

# Reporting for specific materials, systems and methods

We require information from authors about some types of materials, experimental systems and methods used in many studies. Here, indicate whether each material, system or method listed is relevant to your study. If you are not sure if a list item applies to your research, read the appropriate section before selecting a response.

## Materials & experimental systems

| n/a | Involved in the study |
|---|---|
| ☐ | ☒ Antibodies |
| ☒ | ☐ Eukaryotic cell lines |
| ☒ | ☐ Palaeontology and archaeology |
| ☒ | ☐ Animals and other organisms |
| ☐ | ☒ Clinical data |
| ☒ | ☐ Dual use research of concern |

## Methods

| n/a | Involved in the study |
|---|---|
| ☒ | ☐ ChIP-seq |
| ☒ | ☐ Flow cytometry |
| ☒ | ☐ MRI-based neuroimaging |

## Antibodies

| | |
|---|---|
| Antibodies used | The therapeutic antibody used for treatment of patients within the clinical trial was pembrolizumab (Keytruda(R)), provided by Merck through NCI/CTEP. |
| Validation | *Describe the validation of each primary antibody for the species and application, noting any validation statements on the manufacturer's website, relevant citations, antibody profiles in online databases, or data provided in the manuscript.* |

## Clinical data

Policy information about clinical studies

All manuscripts should comply with the ICMJE guidelines for publication of clinical research and a completed CONSORT checklist must be included with all submissions.

| | |
|---|---|
| Clinical trial registration | NCT02775851 |
| Study protocol | The study protocol is provided in the Supplemental Information Files. |
| Data collection | Data was collected at individual sites between July 2017 and May 2024. Data was submitted online through the iMedidate Rave platform, which is uploaded daily into the SWOG SQL database. Enrolling sites were responsible for uploading data for participants enrolled on the trial. Each site followed their site-specific rules for data collection and submission, following the timelines provided in the protocol. |
| Outcomes | A one-sided exact binomial test using the method of Clopper and Pearson was used to test the pCR rate against the null hypothesis. Binary proportions are summarized along with 95% confidence intervals. The method of Kaplan-Meier was used to estimate the distributions of relapse-free survival and overall survival, and the log-log method was used to estimate the corresponding confidence intervals for survival at three years. Melanoma-specific survival was calculated as 1-CIF, where CIF is the cumulative incidence of melanoma-specific deaths estimated non-parametrically using the method of Nelson-Aalen. Death from other causes were treated as competing risks. Melanoma-specific survival confidence interval for survival at three years was calculated using the method recommended by Pintilie 20. All analyses were performed using SAS 9.4 (SAS Institute Inc.) and R v4.3.1 (The R Foundation for Statistical Computing). |

