## [Peer Review File · Nature Cancer]

Neoadjuvant PD-1 blockade in surgically resectable desmoplastic melanoma: cohort A of the phase 2 SWOG S1512 trial

Corresponding Author: Dr Antoni Ribas

This manuscript has been previously reviewed at another journal. This document only contains information relating to versions considered at Nature Cancer.

Version 0:

Reviewer comments:

Reviewer #1

(Remarks to the Author)

Nice trial and paper reporting the high response rate to single agent neoadjuvant PD-1 in desmoplastic melanoma. It is certainly novel and original and should be reported. The authors should take the opportunity to further emphasize the preferential use of single agent PD-1 in desmoplastic both for advanced disease and when considering a neoadjuvant approach, given the increased risk of AEs with dual checkpoint blockade.

Comments:

Can you be clear about the 5 pts who had clinical/radiographical nodes at baseline? Why did only 2 pts have nodal dissection?

Were 2 of the 5 the one who did not undergo surgery and the one who had progression?

What was path response based on in the 3 pts that did not have measurable disease at baseline by imaging or photography? How do you know they had residual disease to be assess for pCR after diagnostic biopsy for desmoplastic melanoma if they did not have any clinical evidence of disease? Involved margins on bx specimen would not be adequate to include pts for path response as these pts often have NED on wide excision.

These 3 statements are overstatements given the small sample size and given the fact that no systemic therapy (adjuvant or neoadjuvant) is confirmed to improve survival in resectable melanoma. The authors should state clearly that surgical resection remains the standard of care for resectable disease. The authors should also include in the discussion that adjuvant/systemic therapy for resectable melanoma does not have a confirmed OS benefit, only RFS, and a careful shared decision making between patient and oncologist is still required, but in those patients also opting for systemic therapy after careful consideration of risks and potential benefit, a neoadjuvant approach may be safe and feasible.

With a high pCR and no known patients having died from progressive desmoplastic 15 333 melanoma in this series, it begs the question of whether surgery is then needed in all patients.

The high pCR 286 rate supports the role of systemic treatment for localized desmoplastic melanoma prior to surgical intervention.

The data provide[s] evidence that neoadjuvant anti-PD-1 therapy provides an alternative to the standard approach of 362 wide surgical excision followed by adjuvant radiation and anti-PD-1 immunotherapy.

Reviewer #2

(Remarks to the Author)

The responses to my and my fellow reviewers' comments have improved the clarity of the manuscript's methods and results. I find the issues I raised to be appropriately addressed in this response.

I will note that the questions of the three reviewers prompted re-review of several items (R1: discrepancies in pCR by central vs local review; R1: pt with MPR < 2mm; R2: premature clinicaltrials.gov results reporting; R2: unknown primary site of disease; R3: outlier patient 10)

While these things happen, collectively so much re-review plus several points of wording/typographical errors may raise some concern that aspects were rushed in this manuscript preparation. Would 3 new reviewers raise different issues that prompt further re-review? If, as the authors suggest in their response, "we have effectively established neoadjuvant pembrolizumab as the new standard-of-care treatment for patients with locally advanced desmoplastic melanoma", these data should be unimpeachable when published in their final format.

Reviewer #3

(Remarks to the Author)

The re-submitted paper is a practice changing paper on the role of neoadjuvant therapy in desmoplastic melanoma. The authors have carefully reviewed and responded to all the prior questions, and this has strengthened the finding that desmoplastic melanoma has a high response rate to immunotherapy. The paper is clearly written and carefully presented. It should be accepted as it will change the management of patients with desmoplastic melanoma.

Version 1:

Reviewer comments:

Reviewer #1

(Remarks to the Author)

The authors have now adequately addressed the comments of all reviewers.

Point by point response to Nature Cancer submission NATCANCER-A19784-T

"Neoadjuvant PD-1 blockade in patients with surgically resectable desmoplastic melanoma (SWOG S1512 cohort A)"

Reviewer #1:

Nice trial and paper reporting the high response rate to single agent neoadjuvant PD-1 in desmoplastic melanoma. It is certainly novel and original and should be reported. The authors should take the opportunity to further emphasize the preferential use of single agent PD-1 in desmoplastic both for advanced disease and when considering a neoadjuvant approach, given the increased risk of AEs with dual checkpoint blockade.

Response: We have modified the conclusions in the Abstract and the Discussion to emphasize the preferential use of single agent anti-PD-1 in patients with desmoplastic melanoma.

Comments:

Can you be clear about the 5 pts who had clinical/radiographical nodes at baseline? Why did only 2 pts have nodal dissection?

Were 2 of the 5 the one who did not undergo surgery and the one who had progression?

Response: Five patients staged as having lymph node positive disease had lymphadenectomies after neoadjuvant pembrolizumab, four of them having pCR and one having two residual involved lymph nodes, detailed in the new **Supplementary Table 1**.

Supplementary Table 1. Surgery and pathology analysis of patients with involved lymph nodes at baseline.

ID	Lesion Locations	AJCC Stage	Surgery and pathology assessment
PT0289	Left lower back, left axillary lymph node	T4aN2cM0	Wide excision and left axillary sentinel lymph node resection 0/5 lymph nodes
PT0560	Right submandibular lymph node	T4aN1bM0	Right parotid and right neck lymphadenectomy 0/20 lymph nodes
PT0707	Right posterior shoulder, right axillary lymph node	T2aN1bM0	Wide excision and right axillary lymphadenectomy 0/20 lymph nodes
PT0771	Left axillary lymph node	T3aN2cM0	Wide excision and left axillary lymphadenectomy 0/36 lymph nodes
PT0785	Right axillary lymph node, right back subcutaneous nodule	T3bN1bM0	Wide excision and right axillary lymphadenectomy 2/30 positive lymph nodes

What was path response based on in the 3 pts that did not have measurable disease at baseline by imaging or photography? How do you know they had residual disease to be assessed for pCR after diagnostic biopsy for desmoplastic melanoma if they did not have any clinical evidence of disease? Involved margins on bx specimen would not be adequate to include pts for path response as these pts often have NED on wide excision.

Response: Patients without measurable disease at baseline were all required to have a biopsy or fine needle aspiration to confirm the presence of desmoplastic before starting on neoadjuvant pembrolizumab.

These 3 statements are overstatements given the small sample size and given the fact that no systemic therapy (adjuvant or neoadjuvant) is confirmed to improve survival in resectable melanoma. The authors should state clearly that surgical resection remains the standard of care for resectable disease. The authors should also include in the discussion that adjuvant/systemic therapy for resectable melanoma does not have a confirmed OS benefit, only RFS, and a careful shared decision making between patient and oncologist is still required, but in those patients also opting for systemic therapy after careful consideration of risks and potential benefit, a neoadjuvant approach may be safe and feasible.

With a high pCR and no known patients having died from progressive desmoplastic melanoma in this series, it begs the question of whether surgery is then needed in all patients.

The high pCR rate supports the role of systemic treatment for localized desmoplastic melanoma prior to surgical intervention.

The data provide[s] evidence that neoadjuvant anti-PD-1 therapy provides an alternative to the standard approach of wide surgical excision followed by adjuvant radiation and anti-PD-1 immunotherapy.

Response: We have modified these three statements following the reviewer's recommendations. In particular, we have deleted the first sentence, added the consideration of toxicities of systemic anti-PD-1 to qualify the second sentence, and reworded the concluding sentence:

~~With a high pCR and no known patients having died from progressive desmoplastic melanoma in this series, it begs the question of whether surgery is then needed in all patients.~~

The high pCR rate supports the role of systemic treatment for localized desmoplastic melanoma prior to surgical intervention, after consideration of the potential toxicities induced by neoadjuvant anti-PD1 therapy.

The data provides evidence that three doses of neoadjuvant single agent anti-PD-1 therapy could be considered before surgery for patients with resectable desmoplastic melanoma.

Reviewer #2:

The responses to my and my fellow reviewers' comments have improved the clarity of the manuscript's methods and results. I find the issues I raised to be appropriately addressed in this response.

I will note that the questions of the three reviewers prompted re-review of several items (R1: discrepancies in pCR by central vs local review; R1: pt with MPR < 2mm; R2: premature clinicaltrials.gov results reporting; R2: unknown primary site of disease; R3: outlier patient 10)

While these things happen, collectively so much re-review plus several points of wording/typographical errors may raise some concern that aspects were rushed in this manuscript preparation. Would 3 new reviewers raise different issues that prompt further re-review? If, as the authors suggest in their response, "we have effectively established neoadjuvant pembrolizumab as the new standard-of-care treatment for patients with locally advanced desmoplastic melanoma", these data should be unimpeachable when published in their final format.

Response: We acknowledge the Reviewer's comments and assure that the data error recognized after the first article submission has been fixed. We agree that with a US cooperative group clinical trial with less resources for monitoring and auditing the data than most industry-sponsored clinical trials, we should be conservative in the final conclusions. Accordingly, we have amended the sentences pointed out by Reviewer #1 and changed the wording of our cover letter regarding the significance of this work.

Reviewer #3:

The re-submitted paper is a practice changing paper on the role of neoadjuvant therapy in desmoplastic melanoma. The authors have carefully reviewed and responded to all the prior questions, and this has strengthened the finding that desmoplastic melanoma has a high response rate to immunotherapy. The paper is clearly written and carefully presented. It should be accepted as it will change the management of patients with desmoplastic melanoma.

Response: We again want to thank the Reviewers for the useful comments and guidance to provide this final version of this clinical trial report.